# GENERALIZED FOURIER FEATURES FOR COORDINATE-BASED LEARNING OF FUNCTIONS ON MANIFOLDS

## ABSTRACT

Recently, positional encoding of input coordinates has been found crucial to enable learning of high-frequency functions with multilayer perceptrons taking low-dimensional coordinate values. In this setting, sinusoids are typically used as a basis for the encoding, which is commonly referred to as "Fourier Features". However, using sinusoids as a basis assumes that the input coordinates lie on Euclidean space. In this work, we generalize positional encoding with Fourier features to non-Euclidean manifolds. We find appropriate bases for positional encoding on manifolds through generalizations of Fourier series. By ensuring the encodings lie on a hypersphere and that the appropriate shifts on the manifold preserve inner-products between encodings, our model approximates convolutions on the manifold, according to the neural tangent kernel (NTK) assumptions. We demonstrate our method on various tasks on different manifolds: 1) learning panoramas on the sphere, 2) learning probability distributions on the rotation manifold, 3) learning neural radiance fields on the product of cube and sphere, and 4) learning light fields represented as the product of spheres.

## 1 INTRODUCTION

Recent breakthroughs on learning representations of 3D shapes (Mescheder et al., 2019; Park et al., 2019; Sitzmann et al., 2019) or scenes (Mildenhall et al., 2020) employ the so-called "coordinate-based" networks, which take low-dimensional coordinates as inputs and approximate a continuous function. These are sometimes called "implicit models", when the approximated function implicitly represents the desired output; a typical example is using a signed distance function to represent a 3D shape (Park et al., 2019).

Perhaps the most important recent advancement in this line of research is NeRF (Mildenhall et al., 2020). One of the reasons for NeRF's impressive performance is the positional encoding of input coordinates using sinusoidals of various frequencies, a technique that has been widely adopted (Liu et al., 2020; Schwarz et al., 2020; Yariv et al., 2020) and studied (Tancik et al., 2020; Zheng et al., 2021). The sinusoidals typically used for positional encoding are elements of orthonormal bases for functions on Euclidean spaces. Our key observation is that, to generalize this idea to non-Euclidean manifolds, we should use orthonormal basis functions on the manifold.

Figure 1 (middle) illustrates the evaluation of the Euclidean basis functions on spherical coordinates, which break orthogonality, results in uneven frequency distribution and singularities near the poles. The spherical harmonics are orthonormal basis functions for the sphere and do not exhibit these undesired properties.

NeRF (Mildenhall et al., 2020) and subsequent works avoid the problem just described by parametrizing the view direction as a unit vector in $\mathbb{R}^3$ instead of two angles. We argue that this overparametrization unnecessarily increases the dimensionality of the problem, since the number of basis functions for a fixed bandwidth $b$ grows exponentially with the number of dimensions $d$. Another relevant example is IPDF (Murphy et al., 2021), which takes inputs on the 3-dimensional manifold $\mathbf{SO}(3)$ in the form of a flattened rotation matrix (a vector in $\mathbb{R}^9$), and applies positional encoding by sparsely sampling the space of $\mathcal{O}(b^9)$ basis elements. We show that our approach improve results for both NeRF and IPDF.

Our basic idea is to find an orthonormal basis for the space of functions on the manifold of interest, and introduce principled methods for choosing a subset of the (possibly infinite) basis elements on which the input coordinates are evaluated. We describe mathematical techniques to obtain bases for large classes of manifolds and show experiments on a variety of them.

Our main contributions are:

• We introduce a principled way to apply positional encoding for coordinate-based learning of high-frequency functions on manifolds.

• We prove that our approach is shift-invariant under the light of the neural tangent kernel (NTK) theory (Jacot et al., 2018; Tancik et al., 2020) and for the appropriate "shift" on the manifold, which implies a manifold convolutional behavior.

• Our experiments show the advantages of the proposed methodology on different applications and manifolds: 1) learning panoramas on the sphere, 2) learning probability distributions on the rotation manifold, 3) learning neural radiance fields on the product of cube and sphere, and 4) learning light fields represented as the product of spheres.

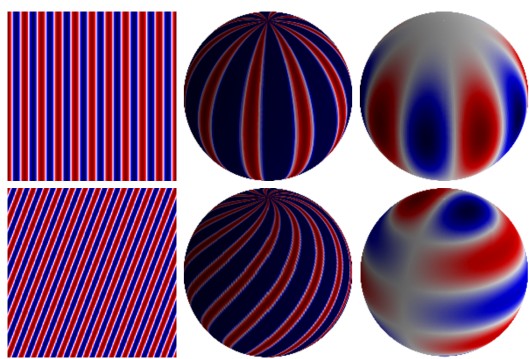

Figure 1: On Euclidean spaces, sinusoids form an orthonormal basis (left) and are used for positional encoding in coordinate-based MLPs. The same functions, when applied to other manifold, are distorted and not orthogonal. We propose to use orthonormal basis functions for the underlying manifold for positional encoding. For the sphere, we use the spherical harmonics basis (right).

## 2 RELATED WORK

The idea of positional encoding using a Fourier basis appeared as early as Rahimi & Recht (2008). They introduced the so-called random Fourier features to accelerate training of kernel methods. The idea is to approximate a shift-invariant kernel with random directions sampled from its Fourier transform, on which the inputs are evaluated. More recently, positional encoding has become popular for sequence modeling in natural language processing and often appears in attention layers and transformers as a way to encode the order of the input tokens. Gehring et al. (2017) use a learned embedding while Vaswani et al. (2017) use the Fourier basis for encoding input token positions. Alternative approaches were introduced by Xu et al. (2019); Wang et al. (2020).

Rahaman et al. (2019) demonstrated that neural networks tend to learn low frequencies more easily than high frequencies, a property called "spectral bias". It also shows, for a simple task of approximating an 1D function, that projecting the input coordinate into a basis of sinusoids facilitates learning high frequencies. NeRF (Mildenhall et al., 2020) reproduced this finding in the challenging task of photorealistic novel view synthesis from a collection of images. They showed that positional encoding is crucial to achieving photorealism, and significantly outperformed the previous state of the art, which included other coordinate-based MLPs (Sitzmann et al., 2019). Concurrently, Zhong et al. (2020) demonstrated the usefulness of Fourier encoding for reconstruction of 3D protein complexes.

Tancik et al. (2020) conducted an in-depth study of positional encoding for coordinate-based MLPs. They show that, under the Neural Tangent Kernel (NTK) theory (Jacot et al., 2018), the positional encoding using a sinusoidal basis results in a stationary kernel, which in turn can be interpreted as a convolutional (shift-invariant) reconstruction filter, desirable for signals on Euclidean spaces. The theory also explains the "spectral bias" (Rahaman et al., 2019) of these models via the rapidly decaying NTK eigenvalues for MLPs. In practice, Tancik et al. (2020) propose randomly sampling the frequencies of the Fourier basis elements for the encoding, and show that it outperforms previous coordinate-based MLPs with no positional encoding (Mescheder et al., 2019), or axis-aligned positional encoding (Mildenhall et al., 2020).

While groundbreaking results and insightful theoretical developments have been demonstrated, most of the attention so far has been focused on coordinate-based MLPs for functions on Euclidean spaces. In this paper, we focus on the non-Euclidean case.

One alternative to Fourier features was introduced by Sitzmann et al. (2020), who use the sine function as the MLP nonlinearity. Major differences are that the sinusoidal is applied on every layer, and the frequencies are defined by the network MLP weights. Zheng et al. (2021) studied the trade-off between memorization and generalization in positional encoding, showing the former is related to the rank of the embedding matrix and the latter is related to the distance preservation of the embedding. They also propose another alternative to Fourier features that consists of sampling a Gaussian at fixed offsets from the input.

## 3 BACKGROUND

NeRF (Mildenhall et al., 2020) recently demonstrated the importance of positional encoding for coordinate-based learning of high frequency functions. For a coordinate $x = [x_1, x_2, x_3] \in \mathbb{R}^3$, the following map was applied before the multilayer perceptron (MLP),

$$x_i \mapsto \{\sin(2^0 \pi x_i), \cos(2^0 \pi x_i), \sin(2^1 \pi x_i), \cos(2^1 \pi x_i), \cdots, \sin(2^L \pi x_i), \cos(2^L \pi x_i)\}. \quad (1)$$

Recall that $\{\sin(m^\top x)\} \cup \{\cos(n^\top x)\}$ with $m, n \in \mathbb{Z}^3$ form an orthonormal basis for functions defined on a compact subset of $\mathbb{R}^3$. So Eq. (1) corresponds to evaluating basis functions of different frequencies at input coordinates; we will refer to this method as "Euclidean encoding". In the particular case of Eq. (1), only axis-aligned, powers-of-two frequencies are used, so the basis for bandwidth $2^L$ is only sparsely sampled. Tancik et al. (2020) found it better to randomly sample the frequency space, which results in basis elements that are not axis aligned, but it is still a sparse sample of the complete basis.

In NeRF (Mildenhall et al., 2020), a subset of the input encodes the view direction. A direction can be associated with a point on the surface of the unit sphere, a non-Euclidean manifold. In order to apply the Euclidean encoding to such input, (Mildenhall et al., 2020) represent the direction as a 3D unit vector instead, which increases the input dimensionality from 2D to 3D, and causes the positional encoding to sample from a set of $\mathcal{O}(b^3)$ basis functions, instead of the minimal $\mathcal{O}(b^2)$ for a bandwidth $b$. In IPDF (Murphy et al., 2021), the input coordinates are flattened $3 \times 3$ rotation matrices, which correspond to points on the 3D rotation manifold $\mathbf{SO}(3)$. The Euclidean encoding then corresponds to sampling basis functions on $\mathbb{R}^9$. This results in sampling from a space of $\mathcal{O}(b^9)$ basis functions instead of $\mathcal{O}(b^3)$.

Our key observation is that positional encoding of input coordinates should reflect the geometry of the underlying manifold. We propose to use the appropriate orthonormal basis functions on the manifold to implement this idea. While for many applications the Euclidean manifold is the correct one, for others it is not, as exemplified above. In the following sections we will show how the appropriate basis can be found for different types of manifolds, how to actually choose a subset of the basis to use in the encoding, and demonstrate experimentally the benefits of using them.

## 4 METHOD

### 4.1 OVERVIEW

We consider a neural network that takes as inputs points on an n-dimensional manifold $\mathcal{M}$. We assume there exists an orthonormal basis $B = \{b_i : \mathcal{M} \to \mathbb{R}\}$ for the space of scalar functions on the manifold $L^2(\mathcal{M})$. Our goal is to apply the map $\gamma : \mathcal{M} \to \mathbb{R}^k$ to positionally encode $x \in \mathcal{M}$ before feeding it to the network, where $k$ is the encoding dimension and

$$\gamma(x) = \{b_j(x) \mid b_j \in B' \subset B\}. \quad (2)$$

The first question that arises is how to find the basis for a given manifold. There is no general solution, but specific solutions exist for some large classes of manifolds. For compact groups (not necessarily abelian), the Peter-Weyl theorem establishes a generalization of the Fourier series that

results in a countable set of orthonormal basis functions. It gives rise to the Wigner-D matrix elements that are used as a basis for $\mathbf{SO}(3)$ in Section 4.3. For locally compact abelian groups, the Pontryagin duality can be used, which is a generalization of the Fourier transform. For Riemannian manifolds, an orthonormal basis can be obtained as eigenfunctions of the Laplace-Beltrami operator. For the sphere $S^2$, these are the spherical harmonics used in Section 4.2.

The neural tangent kernel (NTK) theory (Jacot et al., 2018; Arora et al., 2019; Lee et al., 2019) shows that, under certain conditions, an MLP trained for regression $f \colon \mathbb{R}^n \to \mathbb{R}$, converges to the solution of a kernel regression $f(x) \approx \sum_i w_i y_i k(x, x_i)$, where $k$ is the NTK, $w_i$ the kernel regression weights, and the sum is over the whole dataset. Moreover, when the inputs $x$ have constant norm, the kernel depends only on inner products: $k(x, y) = k'(x^\top y)$. In other words, it is rotation-invariant. Tancik et al. (2020) leverages this theory to explain the success of Euclidean encoding – it transforms the rotation-invariant kernel in a shift-invariant one, which results in approximating a convolution operation over the whole training set for inputs in a flat space.

In this paper, the inputs do not lie on Euclidean space, so the shift-invariance in the sense of translations in Euclidean space is not appropriate. For the manifolds considered, we will seek positional encodings that have constant norm (lie on $S^n$) and such that the natural shift on the manifold also preserves the inner products between positionally encoded inputs. Then, the NTK should approximate a convolution on the manifold. We'll show in the following sections that these properties constrain the selection of basis elements.

## 4.2 THE SPHERE $S^2$

The spherical harmonics are eigenfunctions of the Laplace-Beltrami operator on the sphere $S^2$ and thus form an orthonormal basis for the space of square-integrable functions $L^2(S^2)$. The spherical harmonic of degree $\ell$ and order $m$ is given by

$$Y_m^\ell(\theta, \phi) = \alpha_m^\ell P_m^\ell(\cos\theta)e^{im\phi}, \tag{3}$$

where $\theta$, $\phi$ are the colatitude and longitude angles, $P_m^\ell$ is the associate Legendre polynomial of degree $\ell$ and order $m$, with $-\ell \leq m \leq \ell$. The degree $\ell$ indicates the angular frequency, and $\alpha_m^\ell$ is a normalization constant.

Under the NTK umbrella, an MLP taking points on the sphere encoded as 3D unit vectors will converge to a kernel regression with the kernel depending only on inner products. This is convenient because the natural shift on the sphere is a rotation, which preserves inner products. We want to select a subset of the spherical harmonics for positional encoding that maintains these properties: 1) preserves inner-products, and 2) has constant norm.

**Proposition 1.** *For a given $\ell$, let $Y^\ell = [Y_{-\ell}^\ell, Y_{-\ell+1}^\ell \cdots, Y_\ell^\ell]$ be a vector concatenating all the $2\ell+1$ spherical harmonics of degree $\ell$. Let $Rx$ represent the point $x \in S^2$ rotated by $R \in \mathbf{SO}(3)$. Then, for any $x, y \in S^2$ and $R \in \mathbf{SO}(3)$,*

$$\langle Y^\ell(x), Y^\ell(y) \rangle = \langle Y^\ell(Rx), Y^\ell(Ry) \rangle.$$

*Proof.* See Appendix A.1. □

Since $\left\| Y^\ell(x) \right\| = \sqrt{(2\ell+1)/(4\pi)}$, the vector $Y^\ell(x)$ has the same norm for any $x$, and the NTK depends only on inner products.

These results are easily extended to a concatenation of harmonics of multiple degrees, which suggests a strategy for selecting the basis elements for positional encoding: choose any set of degrees $L \subset \mathbb{N}$ and use all harmonics for the chosen degrees: $B' = \{Y_m^{\ell_i} \mid \ell_i \in L, -\ell_i \leq m \leq \ell_i\}$.

For simplicity, we typically choose a degree $\ell_{\max}$ such that $L = \{\ell \mid 1 \leq \ell \leq \ell_{max}\}$. Note that the maximum degree $\ell$ is the only hyperparameter that needs to be selected for both strategies. In contrast, the Gaussian encoding from Tancik et al. (2020) requires both the encoding size and scale as hyperparameters.

### 4.3 THE ROTATION MANIFOLD $SO(3)$

The rotation manifold $\mathbf{SO}(3)$ is non-abelian and compact. The latter property guarantees a Fourier series for the manifold according to the Peter-Weyl theorem, given by the Wigner-D matrices. The Wigner-D matrices are unitary irreducible representations of $\mathbf{SO}(3)$ and its matrix elements form an orthonormal basis for $L^2(\mathbf{SO}(3))$. The elements for the $(2\ell+1) \times (2\ell+1)$ matrix $D^\ell$ of degree $\ell$ are

$$D^\ell_{m,n}(\alpha, \beta, \gamma) = e^{im\alpha} d^\ell_{m,n}(\beta) e^{in\gamma}, \tag{4}$$

where $\alpha$, $\beta$, $\gamma$ are ZYZ Euler angles representing the rotation, $-\ell \le m,\, n \le \ell$, and $d^\ell_{m,n}(\beta)$ is real and proportional to the Jacobi polynomial $P_k^{(a,b)}(\cos\beta)$, with $k$, $a$, $b$ being functions of $\ell$, $m$, $n$.

Since $D^\ell$ is unitary for all $\ell$, for any set of degrees $L \in \mathbb{N}$, an encoding concatenating all matrix elements as follows has constant norm,

$$B' = \{D^{\ell_i}_{m,n} \mid \ell_i \in L, -\ell_i \le m,\, n \le \ell_i\}. \tag{5}$$

The natural shift on $\mathbf{SO}(3)$ is also a rotation (the group acting on itself). Therefore, similarly to the discussion in Section 4.2, it is desirable that the inner products between encodings are invariant with respect to rotations of the input.

**Proposition 2.** *For a given $\ell$, let $vec(D^\ell)$ be the vectorization of the Wigner-D matrix $D^\ell$. Then, for any $R$, $R_1$, $R_2 \in \mathbf{SO}(3)$,*

$$\langle vec(D^\ell(R_1)), vec(D^\ell(R_2)) \rangle = \langle vec(D^\ell(RR_1)), vec(D^\ell(RR_2)) \rangle.$$

*Proof.* See Appendix A.2 ◻

This also holds for a concatenation of multiple $vec(D^{\ell_i})$, so the map in Eq. (5), besides constant norm, also has rotation-invariant inner products, as desired. In practice, instead of an arbitrary set of degrees, we choose an $\ell_{\max}$ and use all degrees $1 \le \ell \le \ell_{\max}$.

### 4.4 PRODUCT OF SPHERES $S^2 \times S^2$

The coordinates for a point in $S^2 \times S^2$ can be obtained by the concatenation of coordinates on each sphere: $(\theta_1, \phi_1, \theta_2, \phi_2)$. A basis for the product manifold can be obtained by product of the basis functions for the sphere. Thus, we have the following orthonormal basis functions for $L^2(S^2 \times S^2)$,

$$Y^{\ell_1, \ell_2}_{m_1, m_2}(\theta_1, \phi_1, \theta_2, \phi_2) = Y^{\ell_1}_{m1}(\theta_1, \phi_1)\overline{Y^{\ell_2}_{m2}(\theta_2, \phi_2)}, \tag{6}$$

with $Y^\ell_m$ as defined in Eq. (3). We will use the following form interchangeably to reduce notation, $Y^{\ell_1, \ell_2}_{m_1, m_2}(x_1, x_2) = Y^{\ell_1}_{m1}(x_1)\overline{Y^{\ell_2}_{m2}(x_2)}$, where $x_1$, $x_2 \in S^2$.

For a given set of pairs of degrees $L \subset \mathbb{N} \times \mathbb{N}$, we consider the following basis functions, consisting of selecting all possible $(m_i, m_j)$ given $(\ell_i, \ell_j)$,

$$B' = \{Y^{\ell_i, \ell_j}_{m_i, m_j} \mid (\ell_i, \ell_j) \in L, -\ell_i \le m_i \le \ell_i, \text{ and } -\ell_j \le m_j \le \ell_j\}. \tag{7}$$

**Proposition 3.** *For a given pair $(\ell_1, \ell_2)$, let $Y^{\ell_1, \ell_2} = \{Y^{\ell_1, \ell_2}_{m_i, m_j}\}$ be a vector containing all the $(2\ell_1 + 1)(2\ell_2 + 1)$ products of spherical harmonics of degrees $\ell_1$ and $\ell_2$. For any $x_1$, $x_2 \in S^2$, the norm of $Y^{\ell_1, \ell_2}$ is a constant: $\left\|Y^{\ell_1, \ell_2}(x_1, x_2)\right\| = \sqrt{(2\ell_1 + 1)(2\ell_2 + 1)/(4\pi)}$.*

*Proof.* See Appendix A.3. ◻

Since the encodings have constant norm, we are interested in whether the inner product between encodings is preserved under input shift, which would lead to the previously discussed desirable properties that follow from the NTK. For the case of $S^2 \times S^2$, an appropriate shift consists in applying the same rotation to both spheres. For example, in our experiment in Section 5.4, the inputs on $S^2 \times S^2$ represent a pair of points on a sphere, so it makes sense that the appropriate shift should rotate both points.

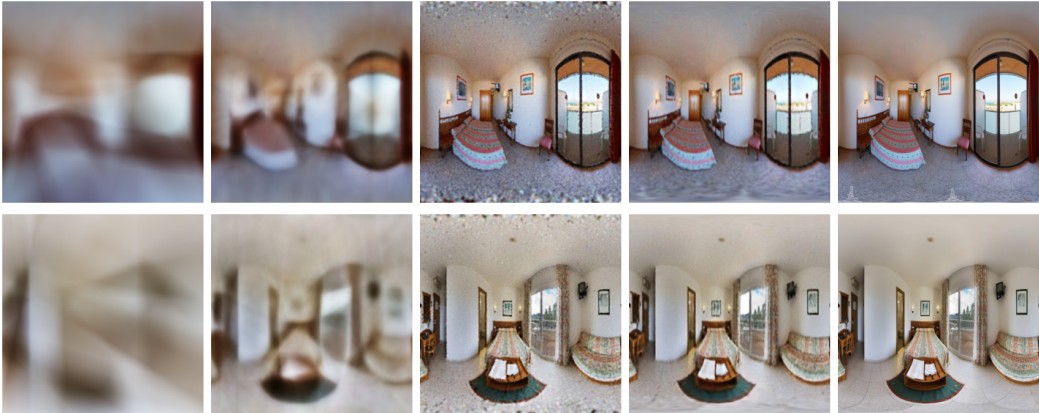

Figure 2: Regressed spherical panoramas. From left to right: no positional encoding, axis-aligned, Gaussian, spherical harmonics (ours) and the ground truth.

**Proposition 4.** *Consider the vector of orthonormal functions $Y^{\ell_1,\ell_2}$ as defined in Proposition 3. The inner product between encodings is invariant to rotations of the inputs:*

$$\langle Y^{\ell_1,\ell_2}(x_1, x_2),\, Y^{\ell_1,\ell_2}(y_1, y_2) \rangle = \langle Y^{\ell_1,\ell_2}(Rx_1, Rx_2),\, Y^{\ell_1,\ell_2}(Ry_1, Ry_2) \rangle,$$

*for any pair $(x_1, x_2) \in S^2 \times S^2$ and $(y_1, y_2) \in S^2 \times S^2$, and rotation $R \in \mathbf{SO}(3)$.*

*Proof.* See Appendix A.4 $\qquad\square$

Therefore, we can construct a basis for positional encoding using Eq. (7). In practice, we set a maximum degree $\ell_{\max}$ and use all pairs under it: $L = \{(\ell_i, \ell_j) \mid 1 \le \ell_i, \ell_j \le \ell_{\max}\}$.

## 5 EXPERIMENTS

### 5.1 SPHERICAL PANORAMAS

Following Tancik et al. (2020); Sitzmann et al. (2020), we evaluate the learning of high-frequencies on a simple task of training an MLP to regress an image from its input coordinates. Since we are interested in non-Euclidean manifolds, we evaluate the methods on 10 spherical panoramas from SUN360 (Xiao et al. (2012)), randomly sampling 1/4 of the pixels as the training set, with probabilities proportional to pixel areas in the input spherical grid.

We follow the image resolution, architecture design, and training schedule of Tancik et al. (2020), and replace their positional encoding with ours, where we use all $(\ell_{\max} + 1)(\ell_{\max} + 2)/2$ spherical harmonics up to $\ell_{\max} = 16$. Table 1 shows the test peak signal-to-noise ratios (PSNRs). Figure 2 depicts the results. Note the large distortions on the poles with Gaussian encoding; these are the areas with large distortion when assuming an Euclidean manifold.

Table 1: Test PSNRs (dB) for spherical panorama regression. We show averages and standard deviations (in parenthesis, referring to last digit) over three runs. for no encoding, axis-aligned (Mildenhall et al., 2020), Gaussian (Tancik et al., 2020), and our spherical harmonics encoding.

|  | No encoding | Axis-aligned | Gaussian | Ours |
|---|---|---|---|---|
| Test PSNR (dB) | 17.91(3) | 20.95(1) | 26.48(7) | 27.08(3) |

### 5.2 PROBABILITY DISTRIBUTIONS ON SO(3)

Recently, Murphy et al. (2021) introduced IPDF, a model that learns to predict probability distributions on $\mathbf{SO}(3)$ to represent the object pose from an image. The main part of the model is a

Table 2: Estimating the pose distribution of symmetric objects on SYMSOL I. DBN refers to "Deep Bingham Networks" Deng et al. (2020), IPDF shows the original results from Murphy et al. (2021), and IPDF* is our modified version. Our model significantly outperforms the baseline in both metrics.

| | Log-likelihood (↑) | | | | | Spread [deg] (↓) | | | | | |
| | avg. | cone | cyl | tet | cube | ico | avg. | cone | cyl | tet | cube | ico |
|---|---|---|---|---|---|---|---|---|---|---|---|---|
| DBN | -1.48 | 0.16 | -0.95 | 0.27 | -4.44 | -2.45 | 22.4 | 10.1 | 15.2 | 16.7 | 40.7 | 29.5 |
| IPDF | 4.10 | 4.45 | 4.26 | 5.7 | 4.81 | 1.28 | 3.96 | 1.4 | 1.4 | 4.6 | 4.0 | 8.4 |
| IPDF* | 5.22 | 6.45 | 5.58 | 6.33 | 6.53 | 1.21 | 4.36 | 1.16 | 1.15 | 5.48 | 4.64 | 9.38 |
| Ours-5 | 5.70 | 6.47 | 5.77 | 6.93 | 6.95 | 2.40 | 2.03 | 1.78 | 1.12 | 2.51 | 1.92 | 2.81 |

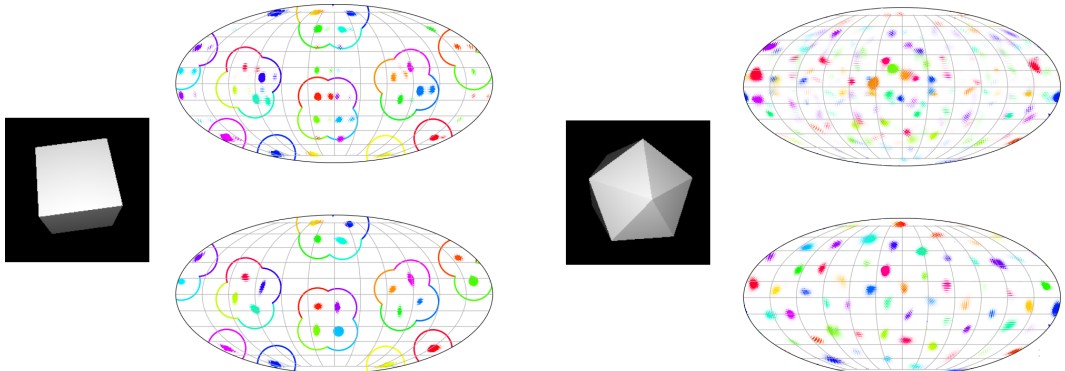

Figure 3: Predicted orientation distributions for the cube and icosahedron. Top row shows "IPDF*" and the bottom shows ours. The baseline has several areas of high probability away from the equivalent ground truths (marked with solid lines), while our method produces a clean distribution. We omit the ground truth annotations for the icosahedron to reduce clutter. Refer to Murphy et al. (2021) for details about the visualization.

coordinate-based MLP taking rotation matrices as inputs. Euclidean positional encoding is applied and shown to improve upon no encoding.

In this experiment, we show that using Wigner-D matrix elements as the basis for the encoding significantly outperforms the Euclidean encoding. We encode the input rotation using all basis elements up to degree $\ell = 5$ by evaluating Eq. (4), using the "Spherical Functions" [1] open source package to perform the computations.

We use the same architecture of Murphy et al. (2021), and train the model on the SYMSOL I dataset, which was introduced by for pose estimation of symmetric solids. We slightly modify the training scheme of IPDF by adding two randomly sampled points near each ground truth and also maximizing the log-likelihood of them both. This stabilizes the training and greatly improves the log-likelihood metric. We denote this modified version "IPDF*", and also apply it to our approach.

Table 2 shows the results. The log-likelihood is the average over all equivalent ground truth poses, and the "spread" metric can be seen as the expected angle error to the closest ground truth (Murphy et al., 2021). Figure 3 compares predictions from our method and the baseline and shows that our approach produces more accurate outputs.

## 5.3 NEURAL RADIANCE FIELDS

NeRF (Mildenhall et al., 2020) has recently shown unprecedented quality on neural rendering and inspired numerous advancements on the field. Part of the success of NeRF is due to the positional encoding of input coordinates. Specifically, the inputs are a point in Euclidean space $x \in \mathbb{R}^3$ and

---

[1]https://github.com/moble/spherical

Table 3: NeRF with spherical encoding on the Blender dataset. "NeRF*" inserts the direction at the 6th layer instead of the 8th. Our method using spherical harmonics for encoding shows small but statistically significant improvements over the baselines. We report PSNRs in dB, with the standard deviations of the last digits shown within parenthesis.

| | avg. | chair | drums | ficus | hotdog | lego | materials | mic | ship |
|---|---|---|---|---|---|---|---|---|---|
| NeRF | 31.73 | 34.08 | 25.01 | 30.51 | 36.84 | 33.52 | 30.19 | 34.43 | 29.27 |
| NeRF* | 31.93(1) | 34.10(5) | 25.32(3) | 30.37(4) | 37.24(3) | 33.29(2) | 31.35(6) | 34.62(7) | 29.17(5) |
| Ours | 32.11(1) | 34.48(6) | 25.41(4) | 30.69(2) | 37.42(9) | 33.29(3) | 31.40(4) | 34.84(4) | 29.37(7) |

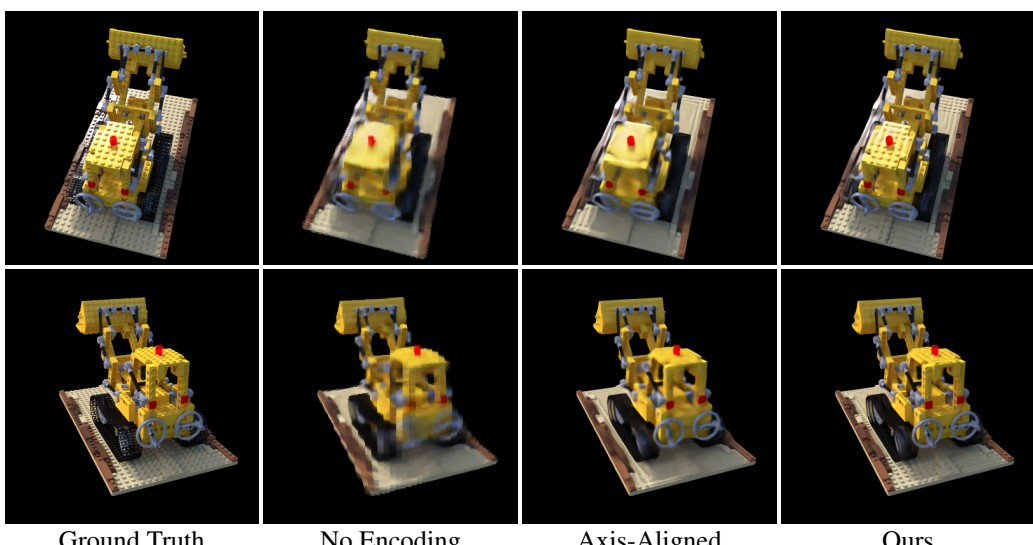

| Ground Truth | No Encoding | Axis-Aligned | Ours |
|---|---|---|---|

Figure 4: Rendering spherical light fields on Lego1.5$k$. Our method using products of spherical harmonics for positional encoding captures high frequency details and produces considerably sharper images as compared to baselines.

a direction $(\theta, \phi) \in S^2$, and the network approximates a density function at $x$, $\sigma(x)$ and the color $c(x, \theta, \phi)$ of the light ray passing through $x$ at direction $(\theta, \phi)$. The Euclidean positional encoding is applied independently to $x$ and to the 3D unit vector $d$ corresponding to $(\theta, \phi)$.

In this experiment, we replace NeRF's Euclidean positional encoding at direction $d$ with an encoding based on the spherical harmonics. We used all the harmonics up to degree $\ell_{\max} = 4$, Evaluating them on device with the `jax.scipy` package (Bradbury et al., 2018).

In NeRF, the directional information is intentionally included only at the last layer, which can be interpreted as a strong Lambertian inductive bias. Therefore the effect of the directional information and its positional encoding in the overall metrics is expected to be small. We noticed that inserting the directional information earlier at the 6th layer instead of 8th slightly increases performance. Our experiments (Table 3) show a small, but consistent and statistically significant improvement when using the spherical harmonics encoding. We repeat each experiment five times and report average PSNRs and standard deviations.

## 5.4 SPHERICAL LIGHT FIELDS

Light field rendering (Levoy & Hanrahan, 1996) is a classic approach for novel view synthesis, which consists of interpolating the radiance at unseen rays based on a large database of given rays, without estimating the scene geometry. Sitzmann et al. (2021) recently combined learning of light fields and coordinate-based MLPs.

By assuming the radiance along a ray is constant, rays can be identified with a 4D vector of coordinates. There are numerous parametrizations in the literature. In this experiment, we consider the

Table 4: Rendering spherical light fields. Axis-aligned refers to the Euclidean positional encoding from Mildenhall et al. (2020) and our method uses product of spherical harmonics.

| PSNR (↑) | | | SSIM (↑) | | | LPIS (↓) | | |
|---|---|---|---|---|---|---|---|---|
| No encoding | Axis-aligned | Ours | No encoding | Axis-aligned | Ours | No encoding | Axis-aligned | Ours |
| 24.94 | 27.42 | 29.14 | 0.82 | 0.87 | 0.91 | 0.16 | 0.13 | 0.10 |

one introduced by Camahort et al. (1998), which represents a ray hitting the scene with the two intersections between ray and a sphere bounding the scene. Therefore, each light ray has coordinates $(\theta_1, \phi_1, \theta_2, \phi_2) \in S^2 \times S^2$.

We introduce a coordinate-based MLP to learn this spherical light field, which enables rendering novel views by querying the ray coordinates for a desired viewpoint. The underlying manifold is a product of spheres, $S^2 \times S^2$, and we use the orthonormal basis for this manifold for positional encoding, as described in Section 4.4. We use all pairs of harmonics up to $\ell_{\max} = 4$.

Rendering the light field directly with an MLP can be orders of magnitude faster than approaches that use ray tracing (for example, NeRF needs to evaluate the MLP more than 100 times to render a single pixel). However, since light fields do not model the underlying scene geometry, there is no inherent notion of multi-view consistency in the functions learned. This hampers a light field rendering models ability to generalize to novel views from few samples. Consequently, akin to classical light field methods, we render a large number of views for training. We sample $1.5k$ HEALPix[2] (Górski et al., 2005; Zonca et al., 2019) points on a hemisphere and render the lego scene from (Mildenhall et al., 2020) with cameras at these points, directed towards the origin.

We train light field models on this dataset using different positional encodings. Table 4 shows the results. We observe an improvement of nearly $1.7$ dB when using the proposed encoding when compared to axis-aligned encoding. Figure 4 shows that our method improves considerably the rendering quality, retaining more high-frequency details as compared to baselines.

## 6 LIMITATIONS

Our method is sensitive to the choice of basis elements. Selecting a maximum frequency that is too high leads to too many elements in the encoding and overfitting, while low frequencies tend to underfit. Similar limitations were observed by Tancik et al. (2020), where the "scale" parameter, which is related to the frequency of the basis elements, needs to be carefully tuned.

Evaluating the special functions that compose the basis used for positional encoding is generally computationally expensive. This is amortized when the inputs lie on a fixed grid (as in the spherical panoramas) or can be pre-computed from the dataset (like in the NeRF (Mildenhall et al., 2020) and light field experiments). This, combined with the fast on-device `jax.scipy` implementation of the spherical harmonics results in almost no overhead for these experiments. For IPDF (Murphy et al., 2021), however, the training grid is sampled randomly, and there is no on-device implementation, which results is half the training speed. During inference the grid is fixed but very large, requiring a 32 Gb device for fast evaluation.

## 7 CONCLUSION

This paper introduced a method to apply positional encoding for learning high-frequency functions on non-Euclidean manifolds. It generalizes the widely used "Fourier Features" by employing orthonormal basis on the manifold for the encoding, as opposed to the sinusoidals that are basis functions for Euclidean spaces. The neural tangent kernel (NTK) theory guides our design; we ensure the encodings lie on a hypersphere and that their inner-products is preserved under the appropriate shift on the manifold. We demonstrated that our approach has advantages with respect to the standard Euclidean encodings in multiple tasks and manifolds.

---

[2]http://healpix.sourceforge.net

ETHICS STATEMENT

We have presented generalized Fourier features for non-Euclidean manifolds. We have intentionally targeted a broad set of applications, and our focus has been on presenting the core mathematical underpinnings of the technique with the objective of making it accessible to a wide audience.

REPRODUCIBILITY STATEMENT

We provide the experimental details necessary for reproducing the results in the appropriate subsections in Section 5 and Appendix B. We note that the implementations are fairly straightforward as in most cases we are simply making small modifications to the positional encoding in existing open source experiment code (e.g. swapping the positional encoding input in NeRF or IPDF). We commit to open sourcing the code for our experiments upon publication.

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

## A PROOFS

### A.1 PROOF OF PROPOSITION 1

*Proof.* A formula for the rotation of spherical harmonics is given by

$$Y^\ell(Rx) = D^\ell(R)^* Y^\ell(x), \tag{8}$$

for all $x \in S^2$ and $R \in \mathbf{SO}(3)$, where $D^\ell$ is a Wigner-D matrix (with elements as defined in Eq. (4)). Since $D^\ell$ is unitary, it preserves inner products. Then

$$\langle Y^\ell(Rx), Y^\ell(Ry) \rangle = \langle D^\ell(R)^* Y^\ell(x), D^\ell(R)^* Y^\ell(y) \rangle = \langle Y^\ell(x), Y^\ell(y) \rangle.$$

□

**Remark** Since $Y_{-m}^{\ell}(x) = (-1)^m \overline{Y_m^{\ell}(x)}$, we can slightly modify the encodings to be real-valued and use only $m \geq 0$ components,

$$
\begin{aligned}
\langle Y^{\ell}(x), Y^{\ell}(y) \rangle &= \sum_{-\ell \leq m \leq \ell} \overline{Y_m^{\ell}(x)} Y_m^{\ell}(y) \\
&= Y_0^{\ell}(x) Y_0^{\ell}(y) + \sum_{-\ell \leq m < 0} \overline{Y_m^{\ell}(x)} Y_m^{\ell}(y) + \sum_{0 < m \leq \ell} \overline{Y_m^{\ell}(x)} Y_m^{\ell}(y) \\
&= Y_0^{\ell}(x) Y_0^{\ell}(y) + \sum_{0 < m \leq \ell} \overline{Y_{-m}^{\ell}(x)} Y_{-m}^{\ell}(y) + \sum_{0 < m \leq \ell} \overline{Y_m^{\ell}(x)} Y_m^{\ell}(y) \\
&= Y_0^{\ell}(x) Y_0^{\ell}(y) + \sum_{0 < m \leq \ell} Y_m^{\ell}(x) \overline{Y_m^{\ell}(y)} + \overline{Y_m^{\ell}(x)} Y_m^{\ell}(y) \\
&= Y_0^{\ell}(x) Y_0^{\ell}(y) + \sum_{0 < m \leq \ell} 2 \Re(Y_m^{\ell}(x)) \Re(Y_m^{\ell}(y)) + 2 \Im(Y_m^{\ell}(x)) \Im(Y_m^{\ell}(y)),
\end{aligned}
$$

where $\Re$ and $\Im$ are the real and imaginary parts, respectively.

This suggests the following alternative encoding, which also has constant norm and rotation-invariant inner products,

$$
B' = \{ Y_0^{\ell_i}, \sqrt{2} \Re(Y_1^{\ell_i}), \sqrt{2} \Im(Y_1^{\ell_i}), \cdots, \sqrt{2} \Re(Y_{\ell_i}^{\ell_i}), \sqrt{2} \Im(Y_{\ell_i}^{\ell_i}) \mid \ell_i \in L \}. \tag{9}
$$

### A.2 PROOF OF PROPOSITION 2

*Proof.* From the representation property of the Wigner-D matrices, we have

$$
D^{\ell}(R R_1) = D^{\ell}(R) D^{\ell}(R_1). \tag{10}
$$

In the following, we use the identity $\text{vec}(B)^* \text{vec}(A) = \text{tr}(A^* B)$, where $x^*$ is the conjugate-transpose of $x$.

$$
\begin{aligned}
\langle \text{vec}(D^{\ell}(R R_1)), \text{vec}(D^{\ell}(R R_2)) \rangle &= \text{vec}(D^{\ell}(R R_1))^* \text{vec}(D^{\ell}(R R_2)) \\
&= \text{tr}(D^{\ell}(R R_2)^* D^{\ell}(R R_1)) \\
&= \text{tr}(D^{\ell}(R_2)^* D^{\ell}(R)^* D^{\ell}(R) D^{\ell}(R_1)) \\
&= \text{tr}(D^{\ell}(R_2)^* D^{\ell}(R_1)) \\
&= \langle \text{vec}(D^{\ell}(R_1)), \text{vec}(D^{\ell}(R_2)) \rangle.
\end{aligned}
$$

$\square$

**Remark** Similarly to the case with the spherical harmonics, in practice we use a real-valued encoding using only $m \leq 0$ basis elements, by leveraging that $D_{m,n}^\ell = (-1)^{m-n} \overline{D_{-m,-n}^\ell}$,

$$
\begin{aligned}
\langle \text{vec}(D^\ell(R_1)), \text{vec}(D^\ell(R_2)) \rangle &= \sum_{-\ell \leq m \leq \ell} \sum_{-\ell \leq n \leq \ell} \overline{D_{m,n}^\ell(R_1)} D_{m,n}^\ell(R_2) \\
&= \overline{D_{0,0}^\ell(R_1)} D_{0,0}^\ell(R_2) + \\
&\quad \sum_{0 < m \leq \ell} \overline{D_{m,0}^\ell(R_1)} D_{m,0}^\ell(R_2) + \overline{D_{-m,0}^\ell(R_1)} D_{-m,0}^\ell(R_2) + \\
&\quad \sum_{0 < n \leq \ell} \overline{D_{0,n}^\ell(R_1)} D_{0,n}^\ell(R_2) + \overline{D_{0,-n}^\ell(R_1)} D_{0,-n}^\ell(R_2) + \\
&\quad \sum_{0 < m \leq \ell} \sum_{0 < n \leq \ell} \overline{D_{m,n}^\ell(R_1)} D_{m,n}^\ell(R_2) + \overline{D_{-m,n}^\ell(R_1)} D_{-m,n}^\ell(R_2) + \\
&\quad \overline{D_{m,-n}^\ell(R_1)} D_{m,-n}^\ell(R_2) + \overline{D_{-m,-n}^\ell(R_1)} D_{-m,-n}^\ell(R_2) \\
&= \overline{D_{0,0}^\ell(R_1)} D_{0,0}^\ell(R_2) + \\
&\quad \sum_{0 < m \leq \ell} 2\Re\left(\overline{D_{m,0}^\ell(R_1)} D_{m,0}^\ell(R_2)\right) + \sum_{0 < n \leq \ell} 2\Re\left(\overline{D_{0,n}^\ell(R_1)} D_{0,n}^\ell(R_2)\right) + \\
&\quad \sum_{0 < m \leq \ell} \sum_{0 < n \leq \ell} 2\Re\left(\overline{D_{m,n}^\ell(R_1)} D_{m,n}^\ell(R_2)\right) + 2\Re\left(\overline{D_{m,-n}^\ell(R_1)} D_{m,-n}^\ell(R_2)\right) \\
&= \sum_{-\ell \leq n \leq \ell} \Re\left(\overline{D_{0,n}^\ell(R_1)} D_{0,n}^\ell(R_2)\right) + \\
&\quad \sum_{0 < m \leq \ell} \sum_{-\ell \leq n \leq \ell} 2\Re\left(\overline{D_{m,n}^\ell(R_1)} D_{m,n}^\ell(R_2)\right).
\end{aligned}
$$

Since $\Re(\overline{a}b) = \Re(a)\Re(b) + \Im(a)\Im(b)$, we obtain an encoding with constant norm and rotation-invariant inner products as follows,

$$
\begin{aligned}
B' = &\{\Re(D_{0,n}^{\ell_i}), \mid \ell_i \in L, -\ell_i \leq n \leq \ell_i\} \cup \\
&\{\Im(D_{0,n}^{\ell_i}), \mid \ell_i \in L, -\ell_i \leq n \leq \ell_i\} \cup \\
&\{\sqrt{2}\Re(D_{m,n}^{\ell_i}), \mid \ell_i \in L, 0 < m \leq \ell_i, -\ell_i \leq n \leq \ell_i\} \cup \\
&\{\sqrt{2}\Im(D_{m,n}^{\ell_i}), \mid \ell_i \in L, 0 < m \leq \ell_i, -\ell_i \leq n \leq \ell_i\}. \quad (11)
\end{aligned}
$$

### A.3 PROOF OF PROPOSITION 3

*Proof.* The elements of $Y^{\ell_1,\ell_2}(x_1, x_2)$ coincide with the matrix elements of $Y^{\ell_1}(x_1) Y^{\ell_2}(x_2)^*$, with $Y^\ell$ as defined in Proposition 1. Let $\|X\|_F$ be the Frobenius norm of $X$. Then,

$$
\begin{aligned}
\left\|Y^{\ell_1,\ell_2}(x_1, x_2)\right\| &= \left\|Y^{\ell_1}(x_1) Y^{\ell_2}(x_2)^*\right\|_F \\
&= \sqrt{\text{tr}((Y^{\ell_1}(x_1) Y^{\ell_2}(x_2)^*)^* Y^{\ell_1}(x_1) Y^{\ell_2}(x_2)^*)} \\
&= \sqrt{\text{tr}(Y^{\ell_2}(x_2) Y^{\ell_1}(x_1)^* Y^{\ell_1}(x_1) Y^{\ell_2}(x_2)^*)} \\
&= \sqrt{\frac{2\ell_1 + 1}{4\pi} \text{tr}(Y^{\ell_2}(x_2) Y^{\ell_2}(x_2)^*)} \\
&= \frac{1}{4\pi}\sqrt{(2\ell_1 + 1)(2\ell_2 + 1)}.
\end{aligned}
$$

$\square$

### A.4 PROOF OF PROPOSITION 4

*Proof.* We write $Y^{\ell_1,\ell_2}$ as the vectorization of the outer product, with $Y^\ell$ as defined in Proposition 1, $Y^{\ell_1,\ell_2}(x_1, x_2) = \text{vec}(Y^{\ell_1}(x_1) Y^{\ell_2}(x_2)^*)$. Then we use the identity $\langle \text{vec}(A), \text{vec}(B) \rangle = \text{tr}(B^*A)$,

the spherical harmonics rotation formula, that the Wigner-D is unitary, and the trace cyclic property,

$$
\begin{aligned}
\langle Y^{\ell_1,\ell_2}(Rx_1, Rx_2), Y^{\ell_1,\ell_2}(Ry_1, Ry_2)\rangle &= \langle \text{vec}(Y^{\ell_1}(Rx_1)Y^{\ell_2}(Rx_2)^*), \text{vec}(Y^{\ell_1}(Ry_1)Y^{\ell_2}(Ry_2)^*)\rangle \\
&= \text{tr}((Y^{\ell_1}(Ry_1)Y^{\ell_2}(Ry_2)^*)^* Y^{\ell_1}(Rx_1)Y^{\ell_2}(Rx_2)^*) \\
&= \text{tr}(Y^{\ell_2}(Ry_2)Y^{\ell_1}(Ry_1)^* Y^{\ell_1}(Rx_1)Y^{\ell_2}(Rx_2)^*) \\
&= \text{tr}(D^{\ell_2}(R)Y^{\ell_2}(y_2)Y^{\ell_1}(y_1)^* D^{\ell_1}(R)^* D^{\ell_1}(R)Y^{\ell_1}(x_1)Y^{\ell_2}(x_2)^* D^{\ell_2}(R)^*) \\
&= \text{tr}(D^{\ell_2}(R)Y^{\ell_2}(y_2)Y^{\ell_1}(y_1)^* Y^{\ell_1}(x_1)Y^{\ell_2}(x_2)^* D^{\ell_2}(R)^*) \\
&= \text{tr}(D^{\ell_2}(R)^* D^{\ell_2}(R)Y^{\ell_2}(y_2)Y^{\ell_1}(y_1)^* Y^{\ell_1}(x_1)Y^{\ell_2}(x_2)^*) \\
&= \text{tr}(Y^{\ell_2}(y_2)Y^{\ell_1}(y_1)^* Y^{\ell_1}(x_1)Y^{\ell_2}(x_2)^*) \\
&= \langle \text{vec}(Y^{\ell_1}(x_1)Y^{\ell_2}(x_2)^*), \text{vec}(Y^{\ell_1}(y_1)Y^{\ell_2}(y_2)^*)\rangle \\
&= \langle Y^{\ell_1,\ell_2}(x_1, x_2), Y^{\ell_1,\ell_2}(y_1, y_2)\rangle.
\end{aligned}
$$

$\square$

## B  EXPERIMENTAL DETAILS

### B.1  SPHERICAL PANORAMAS

**Spherical PSNR**  Since pixels sampled on an equiangular grid on the sphere have different areas, we use a simple quadrature rule to evaluate the PSNR. For co-latitude $\theta$ and longitude $\phi$, the mean squared error (MSE) can be defined as

$$
\text{MSE} = \frac{2\pi^2}{4\pi N_\theta N_\phi} \sum_{\theta_i} \sin(\theta_i) \sum_{\phi_j} [I(\theta_i, \phi_j) - K(\theta_i, \phi_j)]^2 \tag{12}
$$

$$
= \frac{\pi}{2N_\theta N_\phi} \sum_{\theta_i} \sin(\theta_i) \sum_{\phi_j} [I(\theta_i, \phi_j) - K(\theta_i, \phi_j)], \tag{13}
$$

where $I$ represents an image, $K$ is the noisy approximation of $I$, and $N_\theta$ and $N_\phi$ are the number of sampling points along co-latitude and longitude, respectively. The PSNR (in dB) can thus be defined as

$$
\text{PSNR} = 10\log_{10}\left(\frac{\text{MAX}_I^2}{\text{MSE}}\right) \tag{14}
$$

$$
= 20\log_{10}(\text{MAX}_I) - 10\log_{10}(\text{MSE}), \tag{15}
$$

where the maximum possible pixel value of the image $I$ is $\text{MAX}_I = 255$ (each pixel value is represented by eight bits).

**Additional visualizations**  Figure 5 show additional testing output images with both baseline and the proposed encoding methods.

### B.2  SPHERICAL LIGHT FIELDS

For the light field rendering task, we train three models using no encoding, axis-aligned encoding and the proposed product of spherical harmonics encoding. The model is trained to predict the color, given a light field representation of ray in space. The training signal to train the model is computed as the mean-square error between rendered views and ground truth images. We use an MLP with 12 layers, 1024 channels, ReLU activation for intermediate layers and a sigmoid on the output. The models are trained for $250k$ iterations with a batch size of $8192$ using an Adam optimizer with default settings. We use a learning rate schedule with warm-up for $2.5k$ iterations followed by an exponential decay. The intial and final learning rates are set to $2 \times 10^{-3}$ and $2 \times 10^{-5}$ respectively. We set the maximum degree to be 4 for the axis-aligned as well as our encoding.

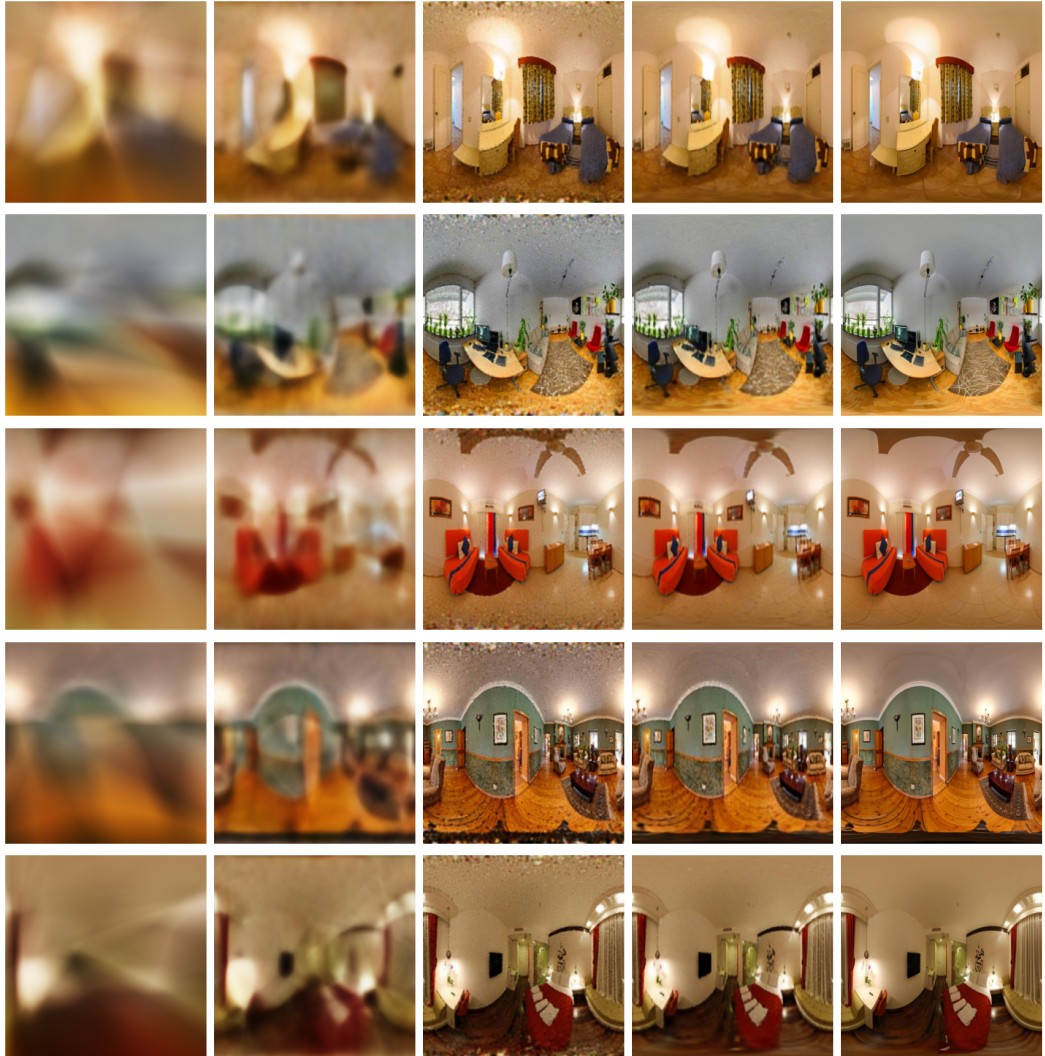

Figure 5: Testing output images(from left to right) with no, Euclidean, Gaussian, and spherical-harmonics encoding and the ground truth (the last column).

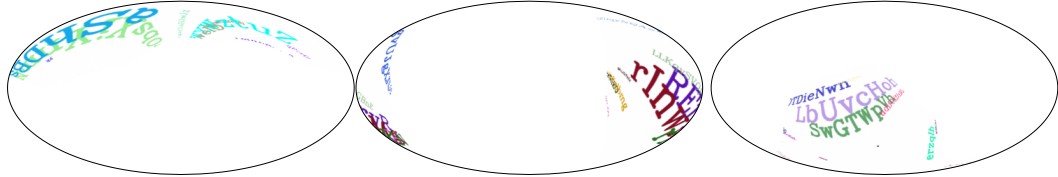

Figure 6: Samples from the "Spherical Text" dataset.

## C    EXTRA EXPERIMENTS

In this section, we show results on a second spherical image dataset for sensitivity analysis to the most important hyperparameters. This dataset is based on the "Text" images of Tancik et al. (2020), which consists of characters of different colors and sizes. It is ideal for investigate positional encoding methods due to the abundance of sharp edges (high frequency content).

We use their 32 original images and perform an stereographic projection to the sphere at random orientations to create spherical images. We refer to this dataset as "Spherical Text".

We also introduce stronger baselines that take 3D points on the sphere, and thus respect the manifold structure, in contrast to the baselines in Section 5.1. The following baselines are considered in this section.

**Axis-aligned 2D**   Similar to the baseline in Section 5.1. Input coordinates are Cartesian, treating the spherical image as a planar image. For a desired encoding size $2M$, and maximum frequency $L$, each coordinate $z_i$ is encoded independently as $z_i \mapsto \{\sin(2\pi \, 2^j z_i)\} \cup \{\cos(2\pi \, 2^j z_i)\}$, where $j = kL/(M-1)$ for $0 \le k < M$ and $i \in 1, 2$. Follows Mildenhall et al. (2020).

**Gaussian 2D**   Similar to the "Axis-aligned 2D", but the tuple of coordinates $z$ is encoded jointly as $z \mapsto \{\sin(2\pi \, b_i^\top z)\} \cup \{\cos(2\pi \, b_i^\top z)\}$, where $\{b_i\}$ are vectors randomly sampled from a centered normal distribution of variance $\sigma^2$ and $0 \le i < M$. Follows Tancik et al. (2020).

**Axis-aligned 3D**   In the 3D case, instead of treating the spherical image as a planar image, we treat it as a sphere embedded in $\mathbb{R}^3$, and take the 3D coordinates of points on the sphere as inputs. The encoding is the same as the 2D case, with $i \in 1, 2, 3$, and encoding size $3M$. This type of encoding is used in the view-direction component in NeRF Mildenhall et al. (2020), which is a spherical function.

**Gaussian 3D**   Similarly to "Axis-aligned 3D", we apply the Gaussian encoding to 3D points on the sphere. The difference with respect to the 2D case is that $b_i$ is $3 \times 1$ instead of $2 \times 1$.

### C.1   BASELINE SENSITIVITY TO FREQUENCY AND SCALE

Table 5: Sensitivity analysis of the maximum frequency $L$ and Gaussian scale $\sigma$ on the "Spherical Text" dataset. Encoding size is fixed to $2M = 512$. The baseline models are highly sensitive to the hyperparameter values. We include our spherical harmonics encoding as reference; for $\ell_{\max} = 22$ the encoding size is slightly lower than $2M$, for $\ell_{\max} = 23$ it is slightly higher.

|  | Freq./Scale | PNSR |
|---|---|---|
| Axis-aligned 3D | $L = 5$ | 25.82 |
| Axis-aligned 3D | $L = 4$ | 26.54 |
| Axis-aligned 3D | $L = 3$ | 25.81 |
| Axis-aligned 2D | $L = 7$ | 25.80 |
| Axis-aligned 2D | $L = 6$ | 26.43 |
| Axis-aligned 2D | $L = 5$ | 26.31 |
| Gaussian 3D | $\sigma = 10$ | 25.99 |
| Gaussian 3D | $\sigma = 8$ | 27.38 |
| Gaussian 3D | $\sigma = 6$ | 28.17(20) |
| Gaussian 3D | $\sigma = 4$ | 28.32(25) |
| Gaussian 3D | $\sigma = 2$ | 27.41 |
| Gaussian 2D | $\sigma = 12$ | 26.84 |
| Gaussian 2D | $\sigma = 10$ | 26.83 |
| Gaussian 2D | $\sigma = 8$ | 26.9 |
| Gaussian 2D | $\sigma = 6$ | 26.82 |
| Ours | $\ell_{\max} = 22$ | 28.45(5) |
| Ours | $\ell_{\max} = 23$ | 28.60(9) |

One downside of all baselines is that there are two hyperparameters to be tuned: the encoding size $2M$ and the scale $\sigma$ or maximum frequency $L$. First, we evaluate the sensitivity of to $\sigma$ and $L$ by keeping the encoding size constant and equal to $2M = 512$. Table 5 shows that the baseline

methods are highly sensitive to these hyperparameter values, which partially explains our method's advantages.

## C.2  SENSITIVITY TO MAXIMUM FREQUENCY

The 2D baselines have the major downside of ignoring the manifold topology. The 3D baselines remove this limitation, but "Axis-aligned 3D" suffers from another limitation. By increasing the dimensions from two to three while keeping the encoding size constant, each dimension must be sampled more coarsely. This problem is exacerbated in IPDF (Murphy et al., 2021), which uses a 9D parametrization of a 3D manifold.

The "Gaussian 3D" baseline mitigates this issue by sampling directions with components on all dimensions. As shown in Table 5, it produces the best results among the baselines, and approaches the performance of our spherical harmonics encoding.

In this section, we thoroughly evaluate the performance of our method against "Gaussian 3D" for different encoding sizes, which determine the maximum frequency of our method. For our spherical harmonics encoding, this is the only hyperparameter to be set, while for the Gaussian we also have to pick the scale $\sigma$. Figure 7 shows that we outperform the baseline for all evaluated encoding sizes.

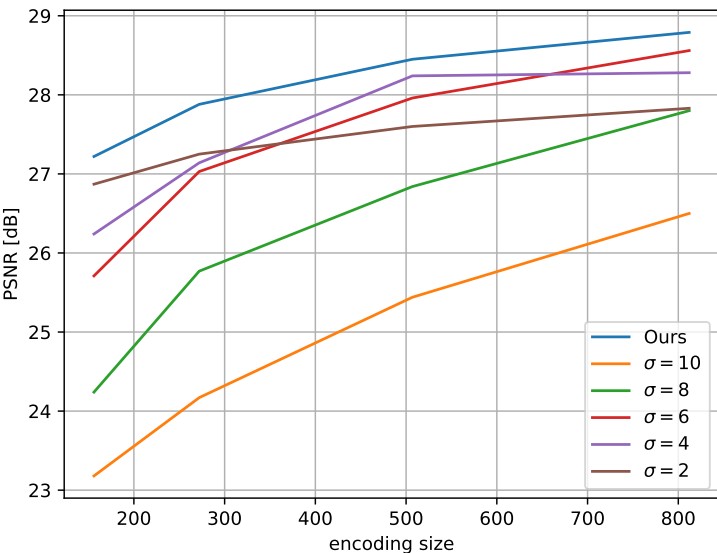

Figure 7: Comparison against "Gaussian 3D" for different sizes of encoding and Gaussian scales, on reconstructing the "Spherical Text" dataset. Our spherical harmonics encoding outperforms the Gaussian Fourier features in all scenarios. The Gaussian encoding best scale depends on the encoding size, which highlights the difficulty to tune these hyperparameters.

## C.3  IMPORTANCE OF SHIFT-INVARIANCE

Another major difference between our method and "Gaussian 3D", is that our encodings have constant norm and rotation-invariant inner products, thus satisfying the NTK conditions for the outputs to approximate a convolution on the manifold (shift-invariance). This is appropriate for the tasks we consider, where the relative importance between training inputs (coordinates) should not depend on their absolute position. The Gaussian encoding does not satisfy these conditions and results in a translation-invariant encoding, which is not appropriate for points on the sphere.

In this section, we show the effects of intentionally breaking these two properties in our model. We sample a vector, fixed during training, from a standard normal distribution with the same dimensions as the encoding. We then replace the positional encoding by its pointwise multiplication with this

vector of random factors. The resulting encoding does not have constant norm nor rotation-invariant inner products.

Table 6 shows the performance on the "Spherical Text" dataset, with and without the random factors. We notice a consistent drop in the average PSNR when $\ell_{max} > 12$.

Table 6: Average accuracy on reconstructing the "Spherical Text" dataset, when random factors are applied to the positional encoding vector, breaking the constant norm and rotation-invariant inner products properties. The performance drops consistently for max degrees above 12.

|  | Max degree | PSNR |
|---|---|---|
| Ours | 12 | 27.40(14) |
| Ours × random factors | 12 | 27.43(17) |
| Ours | 16 | 27.82(11) |
| Ours × random factors | 16 | 27.43(13) |
| Ours | 22 | 28.45(5) |
| Ours × random factors | 22 | 28.13(15) |
| Ours | 28 | 28.79(8) |
| Ours × random factors | 28 | 28.34(8) |

