# OpenReview forum: "Generalized Fourier Features for Coordinate-Based Learning of Functions on Manifolds"
_ICLR.cc/2022/Conference — ICLR 2022 Submitted_

### Official Review · Reviewer_BYCg · 2021-10-31

**Correctness:** 3
**Technical Novelty And Significance:** 3
**Empirical Novelty And Significance:** 2
**Recommendation:** 3
**Confidence:** 4

**Main Review:**

I find the paper is well written, and the experiments seem to be convincing.  However, I have some major concerns.

- First, the author motivated their work (i.e. this paper)  by citing that the original NerF paper used 3-dimensional direction to describe the incoming light direction, hence suffer from over-parameterization. Similarly,  it used the same argument to against the iPDF paper.    This is not true.

More importantly,  it is not clear why achieving "shift-invariance" , "minimal parameterization" or even the orthonomality property matters for the purpose of positional encoding.   The purpose for the common sinusoidal positional encoding is to guide the learning process to move away from the "spectral bias" and to make the learning of high frequency information easier.   Even if the inputs are not parameterized minimally or not shift-invariant, as long as the  basis functions representation contains high frequency fluctuations the above purpose can be met.

 Of course, one may argue that having a proper choice of orthonormal bases would result in some convolution-like behaviour but I don't understand why this is relevant at this stage of positional encoding.   Conversely, one can still use non-orthonormal basis for encoding (e.g. for Fourier features, multiple all  \sin functions by a factor of 2 and leave the \cos functions the same as before, or add identity mapping to basis functions), yet it will not necessarily lead to a worse performance.    As far as the positional encoding is concerned, sometimes having non-minimal redundant and even duplicating information ( lifting) can  be beneficial.   After all, what it matters is whether this can lessen the Spectral Bias of the deep neural network.   The new "theory" developed in this paper is nice to have, but isn't necessarily practically relevant.

Secondly, The paper only includes three  special (and simple) instances non-Euclidean manifolds.   The three example manifolds (sphere, rotation, and product of sphere) are similar and special.  Their natural shifts on manifolds are rotation, and can be almost trivially obtained.   Their Fourier series have been studied by previous work. In other words, these results are well known and not novel.  What about other manifolds, where natural shifts are complex, and Fourier series are unknown ?  For a more generic non-Euclidean manifold it is not clear whether or how a `natural shift' can be defined/derived.   This is though a secondary point.  I am not convinced the work is well motivated, for my first point above.


**Summary Of The Paper:**

The paper generalizes the commonly used sinusoidal position encoding scheme (such as that in NerF input encoding) to inputs naturally residing in non-Euclidean manifold.

This is achieved by representing the input coordinates as  projections on  alternative sets of orthornormal bases instead of the trivial Euclidean coordinates.   The authors argue that by doing so, they extend the translational invariance of Fourier features in Euclidean space to non-Euclidian manifolds, resulting in a convolution-like operator that is invariant to the 'natural shift' on these manifolds.  These new position encoding techniques are tested for several tasks and seem to get consistent improvemnts over their Fourier counterparts.


**Summary Of The Review:**


Strength:
1. This paper presents an interesting idea attempting to extend positional encoding to non-Euclidean manifolds.
2.   This paper provides experiments for three selected manifolds.

Weakness:
The motivation of the proposed work is unclear (see comments above). Moreover, some further arguments are in order.
Experiment provided only show that the selected base functions have similar properties of Euclidean encoding, but that doesn't explain why the selection of base functions can help extract details. Are they really better than  using sinusoids ? Is it really important to design base functions based on the principle? Take sinusoids as example, if we multiply a constant before sine bases and keep cosine bases still, we break the constant-norm property, but the network can still learn from these inputs and mitigate the spectral bias. The increase of performance is not very obvious compared with Euclidean encoding (0.5-1 PSNR increase). Experiments of how basis elements (such as max frequency) affect the performance is not found.

---

> ### Author Response · Authors · 2021-11-23
> **Response to BYCg**
>
> Thank you for the thorough review and helpful comments!
>
> > claims NerF and IPDF suffer from over-parameterization.
>
> Thanks for raising this important point, we realize the wording was not ideal in the paper. Our point is not to argue against overparameterization (after all, the positional encoding that we are proposing is a kind of overparameterization). _Our point is that the axis-aligned encoding employed by NeRF and IPDF have a limitation that, by increasing the number of dimensions while keeping the encoding size constant, each dimension must be sampled more coarsely_. The Gaussian Fourier encoding mitigates this issue by randomly sampling directions with components on all dimensions. We conducted a series of experiments with two extra baselines using the overparameterized 3D representation of points on the sphere: the "Axis-aligned 3D" and "Gaussian 3D". We show that the "Gaussian 3D" outperforms all other baselines, but is still inferior to our approach. Please see our newly written Appendix C for all details.
>
> >  it is not clear why achieving "shift-invariance" , "minimal parameterization" or even the orthonomality property matters for the purpose of positional encoding
>
> The desirable convolutional (shift-invariant) behavior is of the whole model when the task is coordinate-based regression, not particularly for the purpose of positional encoding. Since, according to the NTK results, shift-invariance is achieved given some properties of the inputs, and we are changing the inputs with the proposed encoding, we want to make sure our encoding satisfies those properties.
>
> Inspired by your suggestion regarding (sin, 2cos), we conduct an experiment to verify whether the 1) constant norm and 2) rotation-invariant inner products properties of the encodings matter in practice. The idea is to simply multiply each encoding element by a random factor, and the results show it causes a drop in performance. Please see Appendix C.3 for details.
>
> Please refer to the [top-level](https://openreview.net/forum?id=g6UqpVislvH&noteId=Q8A-1VhahsP) response item 1) for a more detailed motivation.
>
> > proper choice of orthonormal bases would result in some convolution-like behaviour but I don't understand why this is relevant at this stage of positional encoding.
>
> Under the NTK assumptions, the convolution-like behavior arises from the shift-invariance of the inner-product between inputs. So we must have a positional encoding that preserves this property if we wish the regression to approximate convolution on the manifold. For points on the sphere, for example, if we encode the coordinates with sin/cos, the inner products between pairs of encodings will not be rotation-invariant. With our spherical harmonics and other proposed encodings, they are.
>
> > In other words, these results are well known and not novel.
>
> Please refer to the [top-level](https://openreview.net/forum?id=g6UqpVislvH&noteId=Q8A-1VhahsP) response item 2). In summary, we do not claim any new result about harmonic analysis, our contribution is to use those results to derive appropriate forms of positional encoding on manifolds, which has not been done before.
>
> > What about other manifolds, where natural shifts are complex, and Fourier series are unknown. Not clear whether or how a `natural shift' can be defined/derived.
>
> Our method requires computation of an orthonormal basis on the manifold. Please refer to section 4.1 for a discussion of the classes of manifolds that we foresee easy application. Regarding defining a natural shift, there is at least one way for graphs, where the eigenvectors of the graph Laplacian are an orthonormal basis, which is used to define Fourier series and generalized convolutions, which in turn can be used to define a generalized shift.
>
> > Experiments of how basis elements (such as max frequency) affect the performance is not found.
>
> Please refer to the [top-level](https://openreview.net/forum?id=g6UqpVislvH&noteId=Q8A-1VhahsP) response item 3), and the new experiments in Appendix C. In summary, our approach depends on a single hyperparameter which is easily tunable, as opposed to the baselines. Fig. 7 shows how the performance is affected by the encoding size, which depends on the maximum degree.

---

> > ### Comment · Reviewer_BYCg · 2021-11-29
> > **Paper in its current form not ready for ICLR.**
> >
> > Dear Authors,
> >
> > Thank you for your careful responses.  However, I am not convinced.  I stand by my original assessment to this paper ( i.e not good enough, reject),   which is also consistent with the majority of the reviewers' comments.  Most of the reviewers' comments are long, sensible and insightful, and their remarks are meaningful.  Addressing those comments will certainly help to improve the overall quality of this work.

---

### Official Review · Reviewer_8frW · 2021-11-02

**Correctness:** 3
**Technical Novelty And Significance:** 3
**Empirical Novelty And Significance:** 2
**Recommendation:** 6
**Confidence:** 4

**Main Review:**

Strengths:
- For manifold data, it makes sense to use an encoding that adapts to the underlying manifold structure rather than the extrinsic Euclidean coordinates. The submission makes some contribution in this direction;
- Performance improvement verified in the examples;
- The paper discusses multiple application scenarios where the data can be encoded as spheres or their products.

Weakness:
- The title appears to suggest learning on general manifolds, but the actual discussion is focused on manifolds with a spherical structure. Does that mean there will be some manifolds where the proposed approach is not applicable? In particular, does this approach work for manifolds of arbitrary topology? If not, then the title can be slightly misleading and should be revised.
- It is true that the Laplace-Beltrami operator can be used to derive an orthonormal basis on Riemannian manifolds. Unfortunately, there is no further discussion for such cases beyond this general statement. Can the author(s) provide some specific examples of how this should be used?
- The paper confirms that computing the proposed encodings can be expensive. Can the author(s) quantify the increase in computational cost? This should give a more complete picture of the tradeoff between speed and accuracy.

**Summary Of The Paper:**

The paper proposes a positional encoding scheme on manifolds for coordinate-based learning, which achieves better performance than Euclidean coordinate encoding.

**Summary Of The Review:**

The paper makes a contribution towards a more general positional encoding scheme on manifolds. There needs to be more clarification on the applicability of the method on arbitrary manifolds, as well as the computational cost.

---

> ### Author Response · Authors · 2021-11-23
> **Reponse to 8frW**
>
> Thanks for the review and useful comments.
>
> > suggest learning on general manifolds, but the actual discussion is focused on manifolds with a spherical structure
>
> Good point, thanks for raising it. The title is meant to convey that our method is applicable and we demonstrate it in different manifolds, not that it can handle any manifold, as we clarify in section 4.1. Our method works given that we have an orthonormal basis on the manifold. The approach consists in constructing a feature vector for an input coordinate by evaluating the basis elements at that coordinate. For a graph, for example, we would construct a basis using the eigenvectors of the graph Laplacian, and associate to each vertex its corresponding value at each eigenvector.
>
> In our experiments, we have chosen applications for which there are published results with the sinusoidal encoding on non-Euclidean manifolds to clearly demonstrate the effects of using the bases on the manifold themselves through fair comparisons. Photorealistic novel view synthesis tasks have drawn tremendous attention since NeRF (2020) and we find them appropriate to demonstrate our method in practice.
>
> > quantify the increase in computational cost?
>
> We discuss the computational cost in section 6. In summary, for the spherical panoramas, NeRF and light field experiments, all the points queried are known a priori and need to be computed only once. This, combined with the fast on-device jax.scipy spherical harmonics implementation, allows running these experiments with negligible overhead. However, we have no such tool for computing the Wigner-Ds for IPDF, and random points on SO(3) are queried during training so pre-computation is not possible. Our solution is to compute the basis elements on CPU and transfer to the device every step, which makes training 2x slower. This could be greatly improved with an on-device JAX Wigner-D implementation.

---

> > ### Comment · Reviewer_8frW · 2021-11-23
> > **Laplace-Beltrami Comments?**
> >
> > Thank you for the clarification.
> >
> > Could you please comment on my question about the use of Laplace-Beltrami operator for deriving orthonormal basis on Riemannian manifolds? My understanding is that such a construction would require eigendecomposition of a matrix corresponding to the Laplace-Beltrami operator. This could be quite slow even for moderately sized data. Can you provide examples of any practical application of such an approach? Thanks.

---

> > > ### Author Response · Authors · 2021-11-23
> > > **Laplace-Beltrami**
> > >
> > > Thanks for your question.
> > >
> > > It is correct that in general if we are operating on samples of a Riemannian manifold, approximating the eigenfunctions of the Laplace-Beltrami operator with the eigenvectors of the graph Laplacian would be computationally prohibitive when the number of samples is large. However, in our case as in many applications, the full spectrum is not required, and approximations may be suitable. A large body of research exists for approximating spectrum of graphs which could be deployed for large graphs (e.g. Laplacian Eigenmaps [1,2], and Nystrom approximation method [3,4]). These methods have been extended for out-of-sample application useful in our setting where the test time coordinates may not be known during training.   A potential important application that fits this setting can be described as learning coordinate based representations for 3D point clouds, where the observed point cloud is a sampling of a surface better described as a non-Euclidean Riemannian manifold.  The effectiveness a positional encoding constructed using the approximated eigenvectors of the graph Laplacian has many open question and is worth studying in future work.
> > >
> > > Also, on a related note, similar to its application to Euclidean space and the n-Sphere, we mention another example where the Laplace–Beltrami operator can also be explicitly derived is hyperbolic space [5], although the applications are currently outside the scope of the computer vision settings we explored in our paper.
> > >
> > >
> > >
> > > [1] Convergence of Laplacian Eigenmaps, Belkin and Niyogi, 2008. http://people.cs.uchicago.edu/~niyogi/papersps/eigconv_Feb15.pdf
> > >
> > > [2] Laplacian Eigenmaps for Dimensionality Reduction and Data Representation, Belkin and Niyogi, 2003. https://www2.imm.dtu.dk/projects/manifold/Papers/Laplacian.pdf
> > >
> > > [3] Using the Nyström Method to Speed Up Kernel Machines, Williams and Seeger, 2000. https://papers.nips.cc/paper/2000/hash/19de10adbaa1b2ee13f77f679fa1483a-Abstract.html
> > >
> > > [4] Sampling Methods for the Nyström Method, Kumar, Mohri, and Talwalkar, 2012. https://jmlr.org/papers/v13/kumar12a.html
> > >
> > > [5]. Harmonic Analysis on Symmetric Spaces and Applications), Terras, 1985.  https://link.springer.com/book/10.1007/978-1-4612-5128-6

---

### Official Review · Reviewer_1zs2 · 2021-11-06

**Correctness:** 2
**Technical Novelty And Significance:** 2
**Empirical Novelty And Significance:** 2
**Recommendation:** 5
**Confidence:** 4

**Main Review:**

## Strength
The paper provides abundant proof for explaining the key properties of the proposed method. Various empirical study has been provided with illustration to demonstrate the advantage of the method.

## Weakness
1. The neural network model is not very clearly shown though the main claimed contribution is for the positional encoding.
2. The link of the proposed model with the NTK is well described. Actually, there is no big evidence provided by the authors about how the theory and model design are connected with the NTK theory.
3. Since the authors mentioned in Section.6 the importance of selecting an appropriate scale, it would be nice to find complete evidence to support this, i.e., a sensitivity analysis.
The underlying method took degrees up to 10 in the paper. I wonder if this choice is enough for most applications, or the value is only restricted by the computational cost? If its the first case, can the authors provide some justifications on it? Otherwise, how bad influence would it have when the required degree is much larger than the model can provide?
4. In numerical parts, the authors should include more the baselines for comparison.
5. The paper writing could be further improved. Some examples include "positional encoding is crucial to achieve (achieving) photorealism" (Section 2); "A direction can be associated to (with)..." (Section 3).

**Summary Of The Paper:**

This paper generalizes the positional encoding with Fourier features to non-Euclidean manifolds, which has the rotation invariance for feature extraction. The approximates convolutions on the manifold, according to the neural tangent kernel (NTK) assumptions, are shift-invariant.

**Summary Of The Review:**

The theoretical basis of this paper is well supported, but to what degree this method is of practical needs further justification. The whole model of the paper with the proposed positional encoding method is yet to be well stated. The link with the NTK of the proposed method is not evident to readers and may not be the truth as the authors mentioned.

---

> ### Author Response · Authors · 2021-11-23
> **Response to 1zs2**
>
> Thank you for the review and suggestions for improving our paper.
>
> > The neural network model is not very clearly shown
>
> We follow the baseline architectures for experiments 5.1 to 5.3. The architecture is a simple 4 layer, 256 channel MLP for the spherical panoramas (following Tancik et al [1]) and IPDF (following Murphy et al [2]). For the NeRF modification, we follow the original 8 layer MLP architecture (Mildenhall et al [3]), with the minor changes described in 5.3 (denoted NeRF*). For the spherical light fields we use a 12-layer MLP as described in appendix B.2.
>
> > how the theory and model design are connected with the NTK theory
>
> Please refer to the [top-level](https://openreview.net/forum?id=g6UqpVislvH&noteId=Q8A-1VhahsP) response item 1) for more about the role of NTK in our submission. In summary, the NTK theory guides our choice of basis elements such that the learned function approximates a convolution on the underlying manifold.
>
>
> > the importance of selecting an appropriate scale, a sensitivity analysis.
> > should include more the baselines for comparison.
>
> Thanks for the suggestion, these are great points. Please refer to the [top-level](https://openreview.net/forum?id=g6UqpVislvH&noteId=Q8A-1VhahsP) response item 3). In summary, we include new experiments in Appendix C with your suggested sensitivity analysis and comparisons with two stronger baselines.
>
> > The paper writing could be further improved
>
> Thank you for pointing these out. We have corrected the writing.
>
> [1] Tancik et al, "Fourier Features Let Networks Learn High Frequency Functions in Low Dimensional Domains", NeurIPS'20.
>
> [2] Murphy et al, “Implicit-PDF: Non-Parametric Representation of Probability Distributions on the Rotation Manifold.”, ICML’21.
>
> [3] Mildenhall et al, "NeRF: Representing Scenes as Neural Radiance Fields for View Synthesis.", ECCV'20.

---

### Official Review · Reviewer_awek · 2021-11-06

**Correctness:** 4
**Technical Novelty And Significance:** 2
**Empirical Novelty And Significance:** 2
**Recommendation:** 3
**Confidence:** 4

**Main Review:**

1. What is the point of NTK in background section? The authors are not using kernel basis
2. The extensive details about spherical harmonics and Wigner basis can be put into appendix, as it is quite well-known.
3. Same goes for section 4.4, basis for product manifold is well-known as well, so using spherical harmonics as basis for hypersphere, the basis for product of hyperspheres is straightforward. Proposition 3 is well-known result as well.
4. Proposition 4 comes from the invariance property of metric on hypersphere, hypersphere can be seen as a quotient space of rotation group and hence with the canonical metric, proposition 4 is straightforward.
5. The rendering application is section 5.3 is interesting and it is nice to visualize the results by replacing Euclidean encoding with spherical harmonics in NeRF.
6. The authors did not mention anything regarding how to choose the scale of the basis, although they mentioned it as a limitation. Even Papers like "Spherical CNN" by Cohen et al. uses similar basis. It would have been a good contribution if the authors give recipe of choosing maximum bandwidth.

**Summary Of The Paper:**

The authors used coordinate based basis for functions on manifold ans shown different application on sphere and rotation manifold.

**Summary Of The Review:**

The theoretical contribution is not much as most of the method part can be put into appendix and very well-known. The experiments are proof of concept type, and the paper is essentially using well-known basis for functions of manifold, without giving any recipe for choosing frequency or maximum bandwidth L. The contribution is pretty limited as it's essentially using well-known basis in several applications.

---

> ### Author Response · Authors · 2021-11-23
> **Response to awek**
>
> We appreciate you taking the time to review the paper and the constructive criticism.
>
> > What is the point of NTK in background section? The authors are not using kernel basis
>
> Please note that the NTK theory was introduced to analyze the behavior of MLPs in regression tasks, which are the tasks considered in the submission. Please refer to the [top-level](https://openreview.net/forum?id=g6UqpVislvH&noteId=Q8A-1VhahsP) response item 1) for more about the role of NTK in our submission.
>
> > Propositions consist of known results.
>
> Please refer to the [top-level](https://openreview.net/forum?id=g6UqpVislvH&noteId=Q8A-1VhahsP) response item 2). In summary, we do not claim any new result about harmonic analysis, our contribution is to use those results to derive appropriate forms of positional encoding on manifolds, which has not been done before.
>
> > The authors did not mention anything regarding how to choose the scale of the basis
>
> Indeed that was only lightly discussed in the original submission, thanks for flagging. Please refer to the [top-level](https://openreview.net/forum?id=g6UqpVislvH&noteId=Q8A-1VhahsP) response item 3), and the new experiments in Appendix C. In summary, as shown in Fig. 7, our only hyperparameter is easily tunable and typically should be as large as memory permits.

---

### Official Review · Reviewer_CgLb · 2021-11-08

**Correctness:** 4
**Technical Novelty And Significance:** 4
**Empirical Novelty And Significance:** 4
**Recommendation:** 10
**Confidence:** 4

**Main Review:**

This is a nice paper!  Very clearly written, nice background.  I learned alot reading it.

The idea and why you would need to extend Fourier features to non-Euclidean spaces is well motivated and explained.  I appreciate the theoretical approach.

The examples are compelling and give good intuition for why it helps.

Overall an impressive piece of work, I expect others at ICLR will learn from it as well.



**Summary Of The Paper:**

In this paper the authors extend the Fourier features framework, which has recently had great success in 3D scene rendering, to non-Euclidean spaces, including S2, SO(3) and S2xS2.  The results show improvement over prior methods.


**Summary Of The Review:**

A good theoretical approach with results to back it up.

---

> ### Author Response · Authors · 2021-11-23
> **Thank you!**
>
> Thank you so much, we are glad you enjoyed the paper!

---

### Author Response · Authors · 2021-11-23
**Top-level Response**

We thank the reviewers for thoughtful comments and suggestions for improvements. We incorporated suggested changes to the text and introduced new experiments to address reviewer comments in Appendix C.  We address the recurring questions in this top-level answer, and respond to specific points individually.

# 1) NTK, and shift-invariance (reviewers BYCg, awek, 1zs2)
Multiple reviewers asked for clarification on the connection between neural tangent kernel (NTK) theory and our method for generalizing Fourier features to non-Euclidean manifolds.  Earlier works have shown that in the infinite-width limit a fully-connected neural network converges to the kernel regression solution with the NTK (Jacot et al [4]).   Tancik et al [1] exploited this connection for analyzing their proposed Fourier features positional encoding through the properties of the corresponding composed NTK.

One important observation made by Tancik et al [1] is that for many applications a desirable property of the NTK is stationarity (translation invariance) – i.e. the relative importance between inputs (coordinates) should not depend on their absolute positions. Tancik et al [1] showed that their proposed Fourier encoding of Euclidean coordinate inputs maps to a stationary NTK, which is otherwise not the case when operating directly on coordinates without positional encoding.

Similar to the Euclidean setting explored by Tancik et al [1], the desired shift-invariant NTK property guides our methodology for selecting Fourier feature encodings on manifolds.  Specifically, this leads us to restricting our search for positional encodings that have constant norm (hyperspherical) and where the natural shift on the manifold also preserves the inner products between positionally encoded inputs.  We also introduce new experiments in Appendix C.3 demonstrating the importance of shift-invariance.

# 2) Novelty of propositions (reviewers BYCg, awek)

Reviewers commented on the lack of novelty of results presented about spherical harmonics and Wigner-D matrices.

We do not claim to have introduced or discovered any novel results on harmonic analysis or properties of special functions in this paper. Our contribution is to show the benefits of using these well-known bases and properties for positional encoding for coordinate-based learning of functions on manifolds. To the best of our knowledge, this has not been done before.

The point of propositions 1-4 is to show that our chosen set of basis elements satisfy the conditions under which the model output will correspond to a convolution on the manifold according to the NTK theory. As discussed above, this is desirable when learning coordinate-based representations. The choice of basis elements and their factors is not obvious, and we show a theoretical motivation (Sec 4, Appx A) as well as empirical evidence (Sec 5, Appx C) about how to do it.

We believe the context and examples are important for fully understanding the motivations of our method, hence the inclusion of the propositions and brief descriptions of spherical harmonics and Wigner-Ds in the main paper and not in the appendix.

# 3) Sensitivity to hyperparameters  (reviewers BYCg, awek, 1zs2)
Reviewers raised valid points that 1) we did not discuss how to tune the hyperparameters in detail, and 2) did not provide enough results varying such hyperparameters.

One of the advantages of our proposed encodings is that there is a single hyperpameter, the maximum frequency, which is easy to tune.

We have included extra experiments in another spherical image dataset in Appendix C where we 1) compare against two new and stronger baselines (also related to BYCg comments about overparametrization), 2) run a sensitivity analysis on the baselines, showing that their performance is very sensitive to the choice of hyperparameters, and 3) evaluate our model with several values of the single hyperparameter, showing that the performance generally increases with the maximum degree, suggesting that one should use the highest possible degree that fits in memory.

Please see Fig. 7 in Appendix C with the numbers. Note that the encoding size depends on the chosen maximum degree, so there is no other parameter to tune in our model.

[1] Tancik et al, "Fourier Features Let Networks Learn High Frequency Functions in Low Dimensional Domains", NeurIPS'20.

[4] Jacot et al, "Neural Tangent Kernel: Convergence and Generalization in Neural Networks", NeurIPS'18.

---

### Decision · Program_Chairs · 2022-01-20

**Decision:**

Reject

**Comment:**

Positional encoding of the input coordinates using Fourier basis [as described in (1)] is a common tool in the context of multilayer perceptrons (MLP). The author propose to replace the Fourier basis with one on manifolds M (2), such as the classical spherical harmonics (M=S^2), the Fourier basis on M=SO(3) or on M=S^2 x S^2.

MLPs form an important tool of our times. Unfortunately, as it is elaborated by the reviewers the Fourier basis of the investigated manifolds are widely-studied and the presented results are well-known; the submission lacks novelty.